# Are Companies Committed to Preventing Gender Violence against Women? The Role of the Manager's Implicit Resistance

Arístides A. Vara-Horna *, Zaida B. Asencios-Gonzalez, Liliana Quipuzco-Chicata and Alberto Díaz-Rosillo

Facultad de Ciencias Administrativas y Recursos Humanos, Universidad de San Martín de Porres,
Santa Anita 15011, Peru
* Correspondence: avarah@usmp.pe

**Abstract:** This study aims to provide evidence that managers' commitment towards preventing gender violence against women is affected by implicit resistance from the patriarchal culture. A structured questionnaire was given to 673 managers of 243 small, medium, and large private companies in Metropolitan Lima, Peru. We design and test a conceptual model using covariance-based structural equation modeling. Even though 90.3% of managers report being committed to and in favor of preventing gender violence in companies, 48.6% have intense implicit resistance against it. In general, 3 out of 4 managers do not believe in violence against women because they consider it "biased", and think that policies should only talk about family or partner violence. In addition, 2 out of 4 believe that equality policies have "hidden interests" that generate mistrust. The structural equations show that implicit resistance, directly and indirectly, decreases managers' commitment and actions towards preventing gender violence in organizations. Gender biases, irrational beliefs about sexual violence, and a lack of appreciation of gender equality strongly predict these resistances. Business involvement in the prevention of gender violence is a more complex process than expected, requiring a reinforced strategy aimed at overcoming managers' implicit resistance.

**Keywords:** managers; companies; gender; implicit resistance; prevention; violence; women

## 1. Introduction

Gender-based violence against women (VAW) is one of the world's most chronic and prevalent manifestations of gender-based violence. Global estimates report that at least 27 out of every 100 women have been physically or sexually assaulted by their partners or former partners at least once in their relationship (Sardinha et al. 2022). Due to the extent of the problem and its impact on development, the prevention of VAW has become a sustainable development goal that various organizations are urged to follow (Naciones Unidas 2018). Sustainable Development Goal 5 calls for gender equality by eliminating violence and discrimination. It is understood that without the elimination of gender violence and the promotion of gender equality, sustainable development will not be possible. In this context, the International Labor Organization (ILO 2019) calls on companies to join prevention efforts, urging them to create safe spaces free of violence for their workers (Convention 190 and Recommendation 206). The private sector, in effect, is a powerful ally in preventing gender violence. Because companies have resources, power, and influence over their staff, there is increasing interest in involving them.

How are companies involved in the prevention of gender violence? Applying the Maignan and Ralston (2002) model, companies can have three motivations for becoming involved in prevention: value-focused, performance-focused, and stakeholder-focused. 1. Value-driven motivations suggest that a company's ethical or moral commitments might compel its leaders to engage in GBV prevention. This is based on the notion that corporations are not simply legal or economic entities but also moral communities. Many managers believe that human suffering must be prevented, so these ethical standards



can also permeate their thinking (Austin and Wennmann 2017). 2. Performance-based engagement focuses on reducing the costs of GBV. Strategically, prevention is convenient for companies since gender violence also significantly impacts them. Studies in Bolivia, Ecuador, Peru, and Paraguay have found a considerable prevalence of violence against women in companies, translating into labor productivity costs (Vara-Horna 2013, 2016, 2019, 2022). In the meta-analysis carried out by Willness et al. (2007), it was found that workplace sexual harassment (HSL) decreases job satisfaction and organizational commitment and increases turnover intention and physical and mental morbidity. Along the same lines, Au et al. (2022) found that HSL has a highly detrimental effect on the company's value, especially on the performance of future shares. 3. Finally, the involvement based on the stakeholders refers to the demand for the intervention of actors with influence over the companies, which can even be the employees themselves. In this context, preventing gender violence would increase the reputation of the business, consolidating the image of a responsible organization with its staff and society.

Companies are hierarchical organizations and require their leaders' involvement to produce changes in their structure or culture. Therefore, whatever the reason (values, performance, or reputation), preventing gender-based violence in companies requires the managers' commitment (Humbert et al. 2018; Kelan and Wratil 2020; Williamson et al. 2018). In other words, we need to address that motivation towards prevention and translate it into tangible actions. However, this requirement suffers from two problems: 1. Politically correct language that does not reflect real commitment. 2. The emergence of implicit resistance to prevention. Regarding the first problem, management may show an apparent commitment to the growing social demand for gender equality as a form of social desirability. This is possible to the extent that managers are usually people with a high level of education and constant training. Therefore, they can handle a "double discourse" on prevention, but that does not necessarily translate into concrete actions. Regarding the second problem, and again due to their high level of education, managers can develop resistance to prevention in such a way that they justify their inaction or, in the worst case, act contrary to it.

These two problems threaten the global prevention agenda to the extent that they can hinder or delay it, given the illusion of false progress that does not translate into concrete actions and results. Mandatory policies do not guarantee commitment but could activate covert resistance. This resistance to gender is no exception, and has already been reported in the academic literature. Thus, various organizational studies have found that men and women tend to react with higher discrimination and hostility towards women and men who break the conventional stereotype of gender roles at work (Brescoll et al. 2018; Chaney et al. 2019; Fisher et al. 2019; Iacoviello et al. 2021; Infanger et al. 2016; Moss-Racusin et al. 2010; Phelan and Rudman 2010; Rudman et al. 2012a, 2012b; Rudman and Fairchild 2004; Rudman and Phelan 2008; Williams and Tiedens 2016). Women in higher hierarchical positions, women in traditionally masculine workplaces, and labor regulations that repress previously "socially accepted" behaviors can activate gender resistance. On the other hand, research has also suggested that managerial resistance, such as "gender fatigue" or "gender backlash," are significant factors that can explain the lack of progress in organizational gender equity (Colley et al. 2020; Harding et al. 2017; Cortis et al. 2022; Williamson 2019; Thomas and Plaut 2008).

Flood et al. (2021) use the concept of "gender backlash" to explain how men can oppose gender prevention and activism. The "gender backlash" is an adverse, sudden, and violent reaction towards women's empowerment (Alter and Zürn 2020; Faludi 1991; Flood et al. 2021; Mansbridge and Shames 2008). As a form of resistance, gender backlash can manifest itself in many ways: 1. Denial (denies the problem or its legitimacy). 2. Personal denial (refuses to acknowledge responsibility). 3. Inaction (refuses to implement change measures). 4. Appeasement (strives to appease those who advocate change, seeking to limit their impact). 5. Appropriation (pretend to change/support the cause while covertly sabotaging it). 6. Co-option (uses progressive language and goals for reactionary purposes). 7. Repression (reverses or dismantles a change initiative). Thus, according to Flood, denial

seems to be the most common form of resistance, resulting in the denial of the problem, minimizing its scope, meaning, or impact, or redefining its existence. More active conditions such as blame are also frequent: blaming the issue on the victims or reversing the situation by adopting the role of the victim, claiming reverse discrimination, etc. There are also more aggressive ways to discredit the activists' message or credibility, highlighting hidden interests typical of a "conspiracy theory".

However, while backlash is a more obvious form of resistance to women's empowerment, gender fatigue can be more insidious and difficult to identify and address (Williamson 2019). Gender fatigue occurs when gender discrimination exists in the workplace. Still, it is not recognized or denied, resulting in workplaces that appear gender-neutral but are not (Kelan 2009). A simple form of gender fatigue occurs when people are tired of hearing about gender equality, of feeling that they are required to be constantly "politically correct", and of having to attend training sessions on gender and diversity that they believe to be ineffective (Hastings 2011).

From what has been said, it is possible that "gender fatigue" or, even worse, a "gender backlash" is emerging from organizations that could slow down or stop the prevention of gender violence against women. However, there is no evidence in this regard. There is a knowledge gap since no conceptual model identifies this resistance, how it would impact the commitment to prevention, and its corresponding empirical evidence.

### 1.1. Purpose

This study aims to provide evidence that managers' commitment to preventing gender violence against women is affected by implicit resistance from the patriarchal culture, which is still in force. We propose identifying these resistances and their prevalence and impact on managers' commitment to gender violence prevention.

### 1.2. Conceptual Model

Implicit resistance to prevention is a set of ideas, attitudes, and behaviors that adversely affect women's empowerment in organizations. Since managers are usually people with high levels of education and constant training, these resistances are likely implicit; that is to say, they manifest as apparent rationality, hiding behind negative assessments towards the empowerment of women, as well as the presence of high levels of second-generation gender biases and irrational beliefs that justify violence.

Previous research has warned of the existence of second-generation gender biases, which, unlike first-generation gender biases, are neither deliberate nor conscious (Opoku and Williams 2018; Evans and Maley 2021). In this context, "obvious" discriminations are being replaced with less obvious forms of prejudice in companies, becoming prevalent without men and women realizing it is happening (O'Neil and Hopkins 2015; Kolb and McGinn 2008). Second-generation biases are non-conscious and occur when a person continues to make biased evaluations based on stereotypes, despite consciously rejecting them (Orgeira-Crespo et al. 2021). On the other hand, irrational beliefs that justify sexual violence are still widespread in organizations. Blaming the victims of sexual violence, minimizing the facts, and discrediting the complaints because they believe that women have tried to take advantage or have "immoral" behavior are still persistent arguments that reflect the patriarchal socialization of our societies and that tend to decrease the willingness to support the victims (Gramazio et al. 2021; Li and Zheng 2022; Van der Bruggen and Grubb 2014). Both gender biases and justifications towards sexual violence can feed implicit resistance towards prevention.

As has been argued in the literature, achieving gender equality as a sustainable development objective favors the fulfilment of other development objectives (Leal-Filho et al. 2022). For this reason, valuing SDG-5, referring to gender equality as an essential objective, is quite reasonable. However, it is very likely that this appraisal could be limited in unequal contexts, and even more so in men than in women. As such, the subjective value of gender equality may be inversely related to implicit resistance to prevention.

Following the model of Flood et al. (2021), we propose that implicit resistance to prevention is a cognitive–behavioral spectrum that can be organized from the most passive to the most active depending on its intensity. As a result of our experience training managers over the last decade, we have identified four general resistances: two are passive (denial and evasion of responsibility), and the others are active (strategic disarmament and defensive disapproval).

In passive denial, managers can deny the existence of violence against women, arguing equality before the law or cases of assaulted men. Phrases such as "Discrimination against women today does not exist, because we are all equal before the law. Now men and women have the same opportunities", "There should not be the talk of violence against women, but only partner violence, since the Men are also attacked", and "We cannot talk about gender equality if we continue talking only about violence against women. We should only talk about family or partner violence" are typical of this dimension and demonstrate an appropriation of language and legal advances in terms of gender to reverse prevention.

Another form of passive resistance is the avoidance of responsibility. Companies may acknowledge that gender-based violence exists but do not have a direct preventive obligation or believe that companies already do too much. Phrases such as "Companies are not responsible for the existence of these problems of violence. It's the government's responsibility" or "Businesses comply with all laws to prevent sexual harassment or gender discrimination. They should not be asked for more" are characteristics of this dimension.

On the side of active resistance, strategic disarmament is used as an argument focused on effectiveness, sustainability, and priorities: it is not prevented because it does not work, it cannot be sustained, or it is not a priority. Some characteristic phrases are "There are more important priorities in the company than being concerned about some isolated cases of discrimination or gender violence", "I think that training/regulations on discrimination or gender violence are not very useful in companies", and "We have tried to support proposals in favor of women, but many do not make sense or are unsustainable".

Finally, the most intense active resistance is defensive disapproval, where managers can discredit prevention initiatives because they consider them harmful to men or malicious or conspiratorial. Some characteristic phrases are "Talking too much about women demoralizes men at work. They have become the bad guys in the movie", "Now everything is women, violence, and discrimination. I think there is an exaggeration. You must find a balance", and "Many gender equality policies have hidden interests. It's too biased. I do not trust them".

Implicit resistance to prevention antagonizes commitment and preventive actions within companies. In Figure 1, we propose a conceptual model where we explain the impact of these resistances on managers' commitment to prevention and its translation into specific actions within companies.

According to this model, the company's prevention actions depend on the level of the managers' commitment to prevention. A more significant managerial commitment must be translated into more prevention actions; however, this requires managers to value gender equality and reduce their irrational beliefs towards sexual violence and its gender biases. However, in unequal and patriarchal contexts, this resistance would slow any behavior change. The model suggests that in this process, a series of resistances to change will emerge, most of which are implicit (not conscious), with adverse effects on the commitment to prevention. This model also suggests that the impact of these resistances is both direct and indirect. In other words, irrational beliefs, gender biases, and gender assessment will impact the commitment to prevention and prevention actions through implicit resistance. Thus, implicit resistance would also be a mediating variable between attitudes, beliefs, and values on the one hand and commitment and preventive actions on the other.

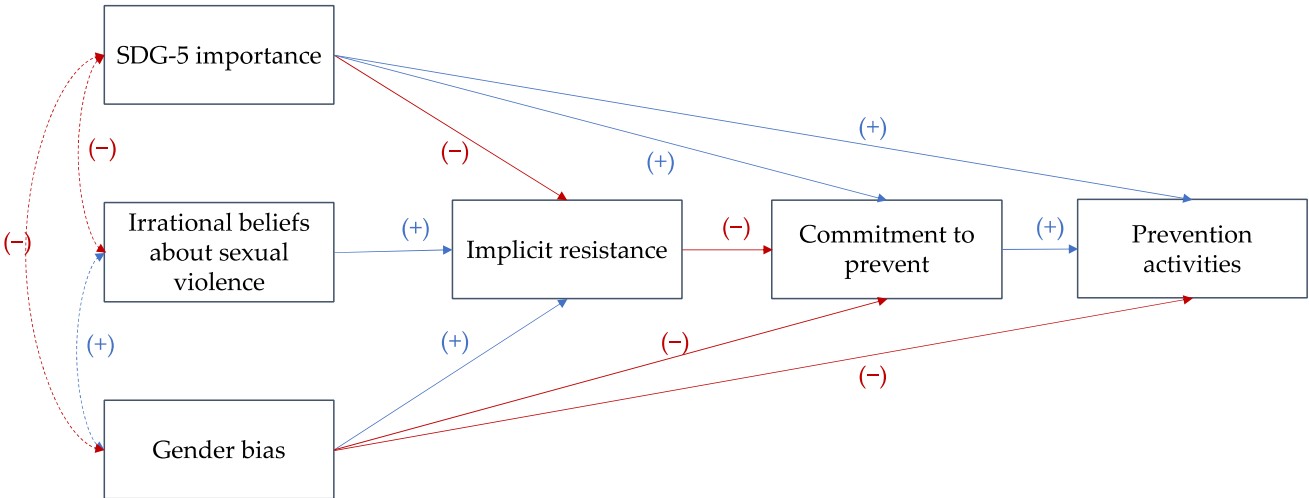

**Figure 1.** Conceptual model of the origin and impact of resistance implicit in managerial commitment and actions to prevent gender violence.

## 2. Materials and Methods

### 2.1. Selection of Sampling Units

To test the conceptual model, we focus on private companies in Peru. Peru is an ideal country to initially probe the model because gender violence is prevalent and deep-rooted; legislation makes its prevention mandatory in companies (at least in part), and increasing numbers of initiatives involve the private sector.

Peru is one of the countries with the highest levels of domestic violence in the region (Bott et al. 2019). According to official figures from the National Observatory on Violence against Women, in 2019, 57.7% of women between the ages of 14 and 49 had been assaulted by their partners (INEI 2022c), and considering women aged 18 and over, this figure becomes 67.6% (INEI 2019). In addition, the National Survey on Social Relations ENARES (INEI 2019) shows that in Peru, there is a high social tolerance for violence against women (58.9%). Gender-based violence against women is also highly prevalent in Peruvian companies. A total of 42.2% of employees of private companies have been assaulted by their partners or ex-partners, and 12.3% report having been attacked by their partners while working (Vara-Horna 2016). Regarding workplace sexual harassment, according to the Elsa report from GenderLab (2021), 34% of workers have experienced some typical manifestation of sexual harassment.

In Peru, companies are obligated to prevent and punish workplace sexual harassment. Law 27942, promulgated in February 2003, establishes that companies must train their personnel in addition to repairing the damage caused and informing the Ministry of Labor. In July 2019, Supreme Decree 014-2019-MIMP was approved, which regulates the law, prescribing annual measurements to identify the problem and yearly training for staff, and reporting and sanction protocols.

Peruvian companies have an increasingly growing body of institutions and organizations that provide services related to preventing gender violence. (1) From the Ministry of Women and Vulnerable Populations, the Safe Company Certification Mark Free of Violence and Discrimination against Women is a recognition given to companies that meet various criteria oriented towards the subject, such as gender mainstreaming, the guarantee of women's labor rights, family–work conciliation, and the prevention-care of gender violence. This recognition began in 2013 and is already in its fifth edition, with increasing coverage. (2) From the private sector, specialized consultancies have emerged in the measurement and prevention of workplace sexual harassment (e.g., GenderLab) or the fight against gender discrimination at work (e.g., Aequales), both with significant acceptance in the sector. (3) Through international cooperation, important initiatives have been developed

to promote the prevention of gender violence in companies. Through its ComVoMujer program, the German cooperation GIZ has developed a whole line of pioneering work on the subject since 2012, being the first to promote the measurement of the business costs of gender violence (Brendel et al. 2017). Spanish cooperation has also promoted important prevention initiatives through the labor insertion of female survivors of gender violence (Vara-Horna 2020).

*2.2. Participants*

This study focused on managers from large, medium, and small companies in Metropolitan Lima. In Peru (until June 2022), there are approximately 3 million formal private companies, 95.4% being micro-enterprises, 3.7% small companies, and 0.6% large and medium-sized companies (INEI 2022a). Private companies located in the provinces of Lima and Callao amount to 1.4 million, demonstrating a high business concentration in the capital (48.8%). In the latter case, 4.7% are small, and 0.9% are large and medium-sized companies. This represents a target population of approximately 73,000 private companies (INEI 2022b).

Through non-probabilistic sampling, 1720 companies from all productive sectors in Metropolitan Lima were contacted and invited to participate in the study. Of these, 243 companies agreed to participate, achieving a response rate of 14.1%. Within each company, apart from the general management, other directive personnel were surveyed. Thus, the participation of 868 managers was achieved, of whom 195 (22.4%) were eliminated because their surveys were incomplete or invalid. Therefore, the final sample amounted to 673 managers with complete and valid surveys. The questionnaire was digitized using SurveyMonkey.

The present study follows a design that complies with the principles of ethics established in The Belmont Report (1979): respect, beneficence, and justice. Regarding the first principle, all participants were informed of the objective and nature of the study and guaranteed informed consent. Regarding non-maleficence, the research sought to preserve the personal safety of the respondents, giving safety priority over information, through anonymous and confidential surveys and without individual access for the participating companies. The companies accessed a global report without identifying who responded in each case. Regarding the third principle of justice, the research results—globally—will be returned to the organizations with specific recommendations and dissemination events.

Regarding the characteristics of the sample, 63.9% are men and 36.1% are women. The mean age is 46.01 years (SD = 12.3). The majority are married or cohabiting (65%), single (23.9%), divorced or separated (9.6%), or widows (1.3%). A total of 71% have children, with an average of 2.02 children (SD = 0.968). Regarding the educational level, almost all of them have completed higher education, be it technical (6.1%), incomplete university (5.7%), university graduate (23%), short postgraduate studies (18.8%), master's studies (38%), or doctorates (7.9%). In terms of position, 13.2% assume the title of Direction or CEO, 34.2% are the management of some specialized area, 28.1% are a boss, 11.6% are a supervisor, and 12.8% have other positions or names. Regarding the specialty of the work, 19.1% are in general management, 11.8% operations/production, 13% human resources, 16.1% sales, commercial, or marketing, 5.7% finance, 6.3% logistics, 4% headquarters management, 1.5% image or public relations, 0.9% social responsibility, and 21.5% other denominations. Regarding seniority in a management position, most have more than eight years in office (32.6%), between 3 to 7 years (35.9%), and less than three years (31.4%). Regarding the sectors, 44.1% work in large companies, 28.2% in medium-sized companies, and 27.6% in small ones. In total, 22.2% of companies have more than 1000 workers, 28% between 100 and 1000 workers, and 49.8% less than 100 workers. Most companies are in the manufacturing (16.8%), services (16.5%), commerce (14%), banking and finance (4.9%), construction (5.1%), mining and hydrocarbons (5.8%), communications (2.2%), transportation (1.5%), agriculture (0.7%), electricity and water (0.6%), fishing (0.4%), and storage (0.7%) industries, among others (30.7%).

*2.3. Instruments*

A structured self-report questionnaire was designed and applied electronically through SurveyMonkey. Apart from demographic and employment information, the questionnaire inquired about the following variables:

Implicit resistance to prevention. The reflective scale of 11 items measures the level of implicit resistance towards preventing gender violence in companies. The scale operationalizes the proposed conceptual model through the measurement of 4 resistance subtypes grouped into two types. Passive denial (3 items) and avoidance of responsibility (2 items) are passive resistances, while strategic disarmament (3 items) and defensive disapproval (3 items) are active resistances. The items are ordinally scaled with six grading points from strongly agree to strongly disagree. The scale is constructed through the average of the items. The presence or absence of implicit resistance is classified depending on whether it exceeds the average of all the items on the scale. Depending on the level of intensity, it is classified as non-intense resistance for a score below three average points, low-intensity resistance for a score between 3 and 4 average points, intense resistance for a score between 4 and 5, and very fierce resistance for a score above five average points.

Commitment to prevention. The reflective scale of 3 items registers the managerial position towards prevention, on a 5-point scale: totally committed, with the possibility of support, neutral, in disagreement, and totally against. Two items focus on the prevention of violence against women in intimate relationships and on workplace sexual harassment, while the third item focuses on the promotion of gender equality. The scale is constructed using the average of the three items.

Gender equality as a value. This is a categorical variable that reflects the personal assessment of management towards gender equality. Of the 17 sustainable development goals, respondents are asked to choose the five most important. If among those chosen is SDG-5, corresponding to gender equality, it is scored 1; otherwise, it is scored 0.

Irrational beliefs about sexual violence. The formative scale of 20 items records irrational beliefs about sexual violence against women in three dimensions: justification (8 items), blame (6 items), and discreditation (6 items). The alternative answers are binary and not mutually exclusive, to the extent that more than one option can be marked as appropriate. This scale was developed by Vara-Horna (2021) in the university context in Ecuador and adapted to the business context in Bolivia (Vara-Horna 2022). One point is assigned to each belief chosen as valid. The scale is built through the sum of the items. The cumulative scale has a significant internal consistency coefficient (McDonald's Omega = 0.730).

Gender biases. The formative scale of 9 items registers the presence of biases to perceive gender barriers. The items have two dimensions: 1. Barriers linked to women (difficulties in getting a job when there are children, workplace sexual harassment, workplace harassment, and domestic violence). 2. Barriers related to work (family–work balance, promotions, job discrimination, achieving high positions, and job security). There is considered to be a gender bias when managers believe that men have more or equal barriers than women due to discrimination or violence. The scale is built through the sum of the items, adding 2 points if the option "men" is chosen or 1 point if "both equally" is chosen. The scale has a significant coefficient of internal consistency (McDonald's Omega = 0.786).

Prevention actions in the company. The formative scale of 18 items is grouped into two sections. The first asks if the company has policies to prevent violence against women and workplace sexual harassment. The second inquiries about all the actions carried out by the company to prevent gender violence. From a list of 16 activities, the managers mark those carried out. The activities include training, internal campaigns, the dissemination of materials, attention to detected cases, the referral of survivors, institutional agreements, external social campaigns, brand creation, community talks, subsidies, and certifications, among others. The scale is built through the sum of the items. The cumulative scale has a significant internal consistency coefficient (McDonald's Omega = 0.870).

Reliability and validity. Confirmatory factor analysis was used to assess the validity of the two reflective measurements. The Chi-square test ($\chi^2$), the RMSEA index, and the SRMR index were used, in which values less than 0.05 indicate a good fit, and values between 0.05 and 0.08 are considered acceptable (Kline 2016). In addition, the comparative fit index (CFI) and the Tucker–Lewis index (TLI) were used, where values greater than 0.95 indicate a good fit, and values greater than 0.90 are considered acceptable (Schumacker and Lomax 2015). The measurement model was evaluated through the internal consistency of Cronbach's alpha coefficient and omega coefficient scales, where a value of $\omega > 0.70$ is appropriate (Raykov and Hancock 2005). Factor loadings ($\lambda$) greater than 0.50 were considered adequate, with the average variance estimate greater than 0.50 for each scale (Hair et al. 2017). As observed in Table 1, the reflective scales present adequate levels of reliability for internal consistency and construct validity.

**Table 1.** Confirmatory factor analysis of the implicit resistance to prevention and commitment to prevention scales.

| Constructs, Dimensions, and Items | Factor Loadings |
|---|---|
| Implicit resistance to prevention (second-order construct) (Average extracted variance = 66.4%/omega reliability = 0.866) | |
| Passive denial (omega reliability = 0.807) | 0.668 |
| 1. Discrimination against women does not exist today since we are all equal before the law. Now men and women have the same opportunities. | 0.589 |
| 2. Violence against women should not be discussed, but only partner violence, since men are also attacked. | 0.838 |
| 3. You cannot talk about gender equality if we only talk about "violence against women". We should only talk about family or partner violence. | 0.843 |
| Avoidance of responsibility (alpha reliability = 0.637) | 0.757 |
| 4. Companies are not responsible for the existence of these problems of violence. It is the government's responsibility. | 0.669 |
| 5. Businesses comply with all laws to prevent sexual harassment or gender discrimination. They should not be asked for more. | 0.763 |
| Strategic disarmament (omega reliability = 0.708) | 0.842 |
| 6. There are more important priorities in the company than being concerned about some isolated cases of discrimination or gender violence. | 0.682 |
| 7. I think that training/regulations on discrimination or gender violence are not very useful in companies | 0.600 |
| 8. We have tried to support proposals in favor of women, but many do not make sense or are unsustainable. | 0.740 |
| Defensive disapproval (omega reliability = 0.837) | 0.965 |
| 9. Talking too much about women demoralizes men at work. They have become "the bad guys in the movie" | 0.765 |
| 10. Now, everything is women, violence, and discrimination. I think there is an exaggeration. It must find a balance. | 0.831 |
| 11. Many gender equality policies have hidden interests. It's too biased. I distrust them | 0.760 |
| Commitment to prevention (Average extracted variance = 70.7%/omega reliability = 0.771) | |
| 1. How committed are you to gender equality in the company? | 0.740 |
| 2. How committed are you to the prevention of workplace sexual harassment? | 0.720 |
| 3. How committed are you to preventing violence against women? | 0.766 |

Note: $X^2$ = 68.136, g.l. = 67, *p* = 0.438; comparative fit index (CFI) = 0.999; Tucker–Lewis index (TLI) = 0.995; goodness of fit index (GFI) = 0.992; root mean square error of approximation (RMSEA) = 0.005, *p* = 0.999; standardized root means square residual (SRMR) = 0.040.

*2.4. Data Analysis Procedure*

To test the existence of significant, direct, and indirect relationships between the variables proposed in the conceptual model, structural covariance equations (SEM) were used, via the Lavaan package in R Studio (Rosseel 2012) and Stata 17. SEM is useful for determining how the independent variables influence the dependent variables. In this sense, the researchers assume that the independent variables can affect other mediating variables, later affecting the dependent variable. Therefore, it is considered that the relationship between the independent and dependent variables is not only direct but may also be indirect (Hair et al. 2017; Hayes 2013; Baron and Kenny 1986). In this case, we use maximum likelihood estimators to identify the precision of the standardized beta coefficients and determine the statistical significance of the hypothesis test; we use robust errors instead of standard errors. This technique estimates the standard error by correcting for heteroscedasticity, which makes it possible to calculate the Z distribution and the *p*-values of the path coefficients. These are considered significant in cases where, for example, <0.05, and when the Z score is greater than the critical value (1.96, 5% significance level).

Another advantage of SEM techniques is that we can test the conceptual model, controlling for the confounding effect of some variables. To select the control variables, we identified those that share variation with the study variables, as observed in Table 2. The number of workers is related to the company's size (Rho = 0.741), so we use only the latter as a control variable. Sex is also significantly associated with SDG-5 and gender bias, so it will also be included as a control. The educational level correlates with almost all the predictors and can be an important confounding variable, so it will also be included in the controls. Age, number of children, and length of service are strongly correlated, so only age will be used as a control to avoid collinearity.

**Table 2.** Correlations between labor and demographic variables with target variables.

|  | ODS-5 Value | Implicit Resistance | Commitment Prevention | Gender Bias | Irrational Beliefs to S.V. | Prevention Activities |
|---|---|---|---|---|---|---|
| Sex (women) | −0.189 ** | 0.095 * | −0.056 | 0.257 ** | 0.108 * | 0.079 * |
| Age | −0.105 ** | 0.070 | −0.103 * | −0.001 | 0.046 | −0.071 |
| Children number | −0.110 ** | 0.073 | −0.075 | 0.023 | 0.102 ** | −0.034 |
| Education level | −0.018 | −0.127 ** | 0.114 ** | −0.126 ** | −0.127 ** | 0.148 ** |
| Seniority in the position | −0.101 ** | 0.117 ** | −0.044 | −0.001 | 0.060 | −0.094 * |
| Number of workers | 0.005 | −0.163 ** | 0.048 | −0.026 | −0.130 ** | 0.329 ** |
| % of female workers | 0.036 | −0.095 * | 0.092 * | −0.053 | −0.075 | 0.173 ** |
| company size | 0.077 * | −0.177 ** | 0.071 | −0.049 | −0.132 ** | 0.366 ** |

** *p* < 0.001; * *p* < 0.05; Rho Spearman.

## 3. Results

*3.1. Managers' Commitment to Prevention*

Most of the managers surveyed report being committed to or in support of the prevention of gender violence (90.3%). Only 7.3% report a neutral position, and 2.4% a position against it. The commitment is relatively less towards the promotion of gender equality than towards the prevention of gender violence (see Table 3).

**Table 3.** Managers' commitment to preventing gender violence in companies (%).

|  | Commitment | Support | Neutral | Disagree | Against |
|---|---|---|---|---|---|
| Promotion of gender equality | 49.5 | 34.0 | 11.9 | 1.0 | 3.5 |
| Prevention of sexual harassment | 61.7 | 33.0 | 4.3 | 0.3 | 0.6 |
| Prevention of intimate partner violence against women | 57.5 | 35.2 | 5.8 | 1.0 | 0.5 |

This commitment is associated with prevention activities. In total, 90.6% of managers report that their company has taken at least some action to prevent gender violence. A total of 81.3% report that their company has established policies to prevent workplace sexual harassment against women, and 53.1% report that guidelines have been set to avoid violence against women. Among the most prevalent prevention actions are training, internal campaigns, the dissemination of informative materials, and attention to detected cases. However, the least frequent are those linked to inter-institutional collaboration and strategic prevention actions, such as creating brands/products with prevention messages. In order of prevalence, these are the training of personnel via human resources or the legal department (45.8%), internal preventive campaigns (45.6%), training for managers (45.3%), the dissemination of informative materials within the company (44.4%), the training of the personnel of the company (43.2%), the care of detected cases (33.6%), timely referral to specialized areas of the company (20.2%), community prevention talks (17.5%), advertising campaigns for the prevention of gender violence (17.5%), timely referral to specialized entities outside the company (11.6%), preventive external campaigns (11%), agreements with external institutions (9.2%), the creation of brands/products with a prevention message (5.6%), the certification of the Safe Company Seal of the Ministry of Women (4.3%), financial support to organizations to prevent violence against women (1.9%), and other prevention actions (26.2%).

### 3.2. Managers' Implicit Resistance

These optimistic results, however, contrast with the prevalence of implicit resistance. In fact, 48.6% of managers have implicit resistance towards preventing gender violence. Figure 2 shows a linear association between both variables: as implicit resistance intensifies, commitment to prevention decreases.

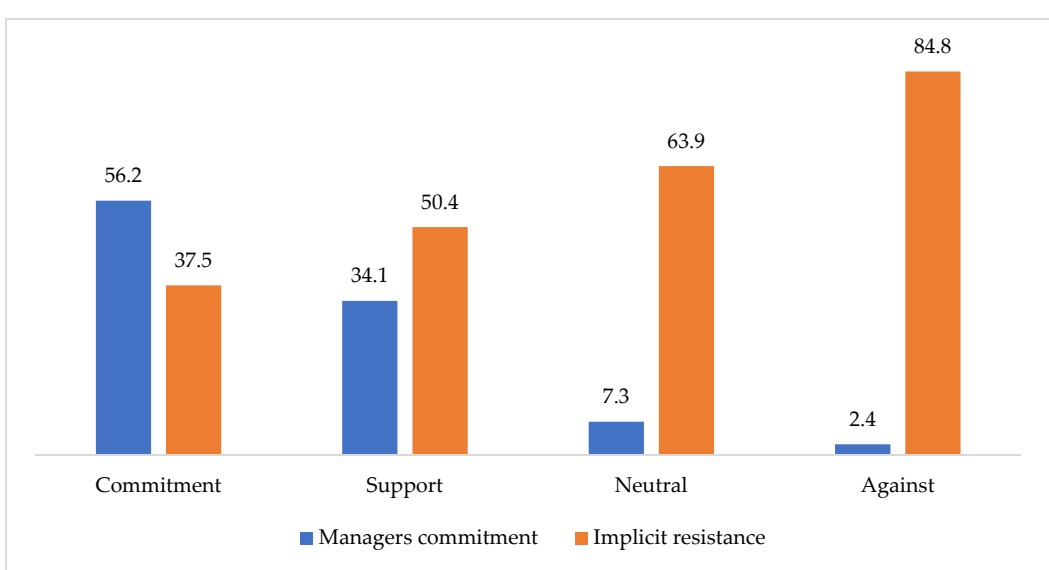

**Figure 2.** Prevalence of manager's commitment and implicit resistance to prevent gender violence in organizations (%). Significant association between both categorical variables (likelihood ratio = 37.999, d.f. = 3, $p < 0.001$).

Almost 100% of managers agree with at least some indicator of implicit resistance to prevention. Passive implicit resistances are more prevalent than active ones. Passive denial resistances are very frequent (between 47.3% and 72.5%), while avoidance resistances are relatively scarce (between 25.3% and 30.9%). Regarding active resistance, defensive action is more prevalent (between 29.4% and 50.4%), followed by strategic disarmament resistance (between 18.7% and 24.6%). Most managers do not believe in violence against women because they consider it "biased" and feel that only family or partner violence should

be talked about. In addition, more than half believe that equality policies have "hidden interests" that generate mistrust (see Table 4).

**Table 4.** Managers' implicit resistance to gender violence prevention in companies (%).

|  | TA | SA | A | D | SD | TD |
|---|---|---|---|---|---|---|
| **Passive denial** | | | | | | |
| Discrimination against women does not exist today since we are all equal before the law. Now men and women have the same opportunities. | 20.5 | 8.9 | 17.9 | 36.3 | 7.9 | 8.5 |
| Violence against women should not be discussed, but only partner violence, since men are also attacked. | 21.2 | 11 | 38.2 | 18.1 | 4.9 | 6.6 |
| You cannot talk about gender equality if we only talk about "violence against women". We should only talk about family or partner violence. | 22.8 | 14.1 | 35.6 | 17.6 | 3.4 | 6.5 |
| **Avoidance of responsibility** | | | | | | |
| Companies are not responsible for the existence of these problems of violence. It is the government's responsibility. | 5.7 | 3.9 | 15.7 | 49.4 | 12.1 | 13.3 |
| Businesses comply with all laws to prevent sexual harassment or gender discrimination. They should not be asked for more. | 5.3 | 5.2 | 20.4 | 50.7 | 9.1 | 9.4 |
| **Strategic disarmament** | | | | | | |
| There are more important priorities in the company than being concerned about some isolated cases of discrimination or gender violence. | 3.9 | 3.3 | 11.5 | 44.7 | 15.8 | 20.9 |
| I think that training/regulations on discrimination or gender violence are not very useful in companies | 3.6 | 3.4 | 17.6 | 43.8 | 15.1 | 16.5 |
| We have tried to support proposals in favor of women, but many do not make sense or are unsustainable. | 4.0 | 7.4 | 27.5 | 41.7 | 9.5 | 9.8 |
| **Defensive disapproval** | | | | | | |
| Talking too much about women demoralizes men at work. They have become "the bad guys in the movie" | 4.9 | 5.2 | 19.3 | 48.2 | 9.5 | 12.8 |
| Now, everything is women, violence, and discrimination. I think there is an exaggeration. It must find a balance. | 8.5 | 7.7 | 34.2 | 31.5 | 9.2 | 8.9 |
| Many gender equality policies have hidden interests. It's too biased. I distrust them | 8.3 | 7.3 | 34.6 | 34 | 7.4 | 8.3 |

TA = totally agree; SA = strongly agree; A = agree; D = disagree; SD = strongly disagree; TD = totally disagree.

### 3.3. Gender Bias and Irrational Beliefs about Sexual Violence

Gender biases have a considerable prevalence. In total, 75.8% of managers report at least some indicator of unperceived gender barriers. As seen in Table 5, there are fewer gender biases linked to women than to the organization. In general, increasing numbers of managers are recognizing that women suffer more gender violence and employment difficulties due to care obligations. However, that group decreases to more than half when gender barriers within the organization must be recognized.

**Table 5.** Unperceived gender barriers (gender bias) (%).

| Gender Barriers (Biases) | Men | Both | Women |
| --- | --- | --- | --- |
| Difficulties getting a job when they have young children | 0.8 | 9.5 | 89.7 |
| Sexual harassment at work | 0.3 | 9.1 | 90.6 |
| Suffer more workplace bullying | 0.6 | 16.8 | 82.6 |
| Suffer more domestic violence | 0.5 | 17.3 | 82.3 |
| Difficulties reconciling family and work life | 4.4 | 33.3 | 62.3 |
| Challenges in achieving high positions in companies | 3.6 | 37.2 | 59.2 |
| Discrimination at work | 1.4 | 40.5 | 58.2 |
| Difficulties getting promotions | 2.3 | 45.2 | 52.5 |
| Difficulties feeling safe at work | 3.3 | 47.4 | 49.3 |

Irrational beliefs towards sexual violence are also prevalent. In total, 53.6% of the surveyed managers report at least some indicator of such beliefs. The most pervasive irrational beliefs are those associated with an imbalance of power in favor of women and to the detriment of men. In this regard, 1 in 3 managers believe that women sexually arouse men to take advantage of the job, then retract and appear to be harassed. For this reason, 27 out of 100 believe that the laws do not protect men from false accusations, and 14 out of 100 believe that many complaints are false because women want attention or revenge. Managers mistakenly believe that sexually assaulted women are responsible for what happened: "When she sexually arouses men to get favors or take advantage. She turns him on, and then she pulls back" (35.7%); "Laws do not protect men from false accusations" (27%); "When the woman allowed the man to be alone with her in intimate situations" (19.2%); "When she starts flirting and then does not measure the consequences" (14.9%); "Many complaints are false because women want attention or revenge for harm" (14.3%); "Many allegations are exaggerations" (8.6%); "When the woman did not ask for help or denounced, if she did not protest, it was because she agreed. She is slow to report" (8%); "When she is exposed to dangerous situations, such as dressing very provocatively or giving a lot of confidence" (7.3%); "When the woman is promiscuous or has a bad reputation, she cannot be trusted with her word" (7%); and "When the woman has already had a relationship with that person before" (6.4%).

*3.4. Gender Differences*

As can be seen in Table 6, women have less gender bias than men and value SDG-5, "gender equality", more. A total of 36.8% respondents chose SDG-5, "gender equality", as an essential variable. While only 30% of male managers consider SDG-5 as necessary, 49% of female managers have chosen it. Similarly, while 64.7% of female managers report some gender bias, 82.0% of male managers do. Finally, while 46.5% of female managers possess some irrational beliefs towards sexual violence, 57.7% of all managers do. However, these differences do not translate into less implicit resistance or more commitment to prevention. Although there is a trend in favor of women, the values are statistically similar in both cases. While 89.0% of female managers affirm that they are committed to the prevention, 83.6% of all managers affirm likewise. Regarding implicit resistance, while 46.5% of women managers have some resistance towards prevention, 49.8% of all managers do.

**Table 6.** Gender differences.

| | B | S.E. | Wald | Sig. | 95% Confidence Interval (Exp(B)) | |
| --- | --- | --- | --- | --- | --- | --- |
| | | | | | Lower | Upper |
| Gender bias | −1.998 | 0.371 | 28.947 | <0.001 *** | 0.065 | 0.281 |
| Irrational beliefs about sexual violence | −0.629 | 0.712 | 0.782 | 0.377 | 0.132 | 2.150 |
| SDG-5 importance | 0.658 | 0.186 | 12.545 | <0.001 *** | 1.342 | 2.780 |
| Implicit resistance | −0.016 | 0.125 | 0.016 | 0.898 | 0.795 | 1.299 |
| Commitment to prevent | −0.230 | 0.153 | 2.248 | 0.134 | 0.589 | 1.073 |

Note. Logistic regression. Maximum likelihood estimator. *** Statistical significance at 99.9%.

### 3.5. The Role of Implicit Managerial Resistance

As seen in Figure 3, and according to the proposed conceptual model, implicit managerial resistance to prevention significantly reduces commitment to prevention and prevention actions in the company. In addition, this resistance is explained by gender biases, irrational misogynist beliefs, and a low appreciation of SDG-5, "gender equality". As resistance intensifies, the gaps between these variables become larger.

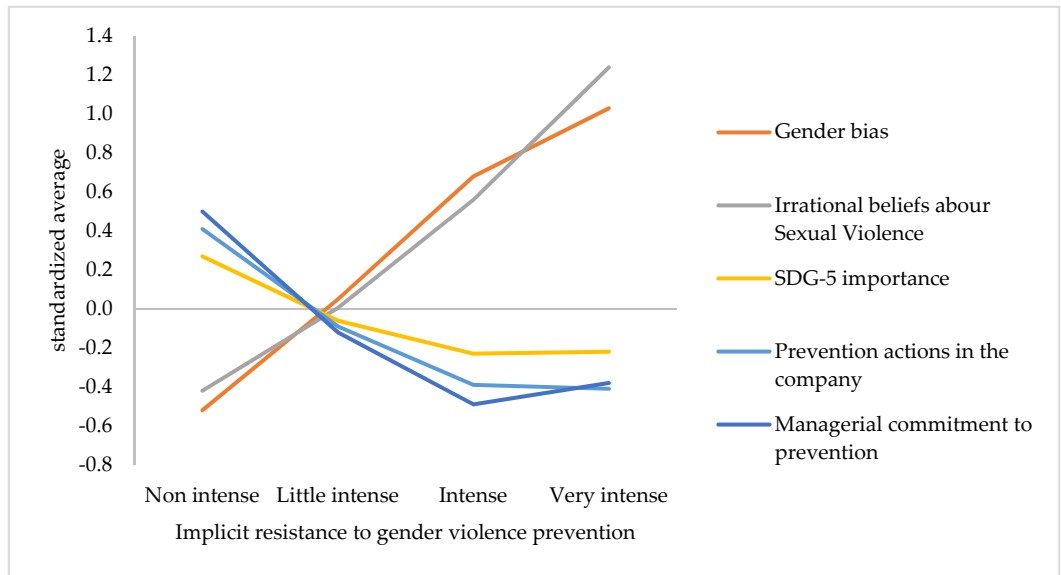

**Figure 3.** Managerial profile towards the prevention of violence against women according to the level of implicit resistance. Note: Standardized mean difference. Gender biases (F = 46.251, $p <$ 0.001), irrational beliefs about sexual violence (F = 37.894, $p <$ 0.001), SDG-5 importance (F = 7.471, $p <$ 0.001), personal commitment towards prevention (F = 27.732, $p <$ 0.001), and prevention actions in the company (F = 18.903, $p <$ 0.001).

Indeed, as seen in Table 7, the implicit resistance towards prevention is powerfully charged by gender biases and irrational beliefs towards sexual violence. Both variables are significant predictors and increase resistance. Conversely, the assessment of SDG-5 "gender equality" reduces implicit resistance to prevention. The control variables have no effect; the implicit resistance towards prevention is the same in both men and women, regardless of their age, educational level, and company size. On the other hand, the commitment to the prevention of gender violence increases when managers value SDG-5 and when they have a higher educational level. However, it decreases when there are gender biases and an implicit resistance to prevention. Indirectly, these effects are enhanced through implicit resistance, including irrational beliefs towards sexual violence. Finally, the company's prevention actions are strongly associated with the size of the company, being

more prevalent in the most prominent companies, albeit becoming less commonplace as age increases. Still, there is also a significant direct impact of the commitment to prevention. Unfortunately, implicit resistance, gender biases, and irrational beliefs towards sexual violence play an important negative role in reducing prevention actions in companies. Finally, the protective role of the assessment of SDG-5 is maintained indirectly.

**Table 7.** Effects of implicit resistance on the commitment and actions to prevent gender violence in companies, controlling for sex, age, educational level, and company size.

| | Standardized Effect (Beta) | | |
| --- | --- | --- | --- |
| | **Direct** | **Indirect** | **Total** |
| **Implicit resistance** | | | |
| Gender bias | 0.315 *** | – | 0.315 *** |
| Irrational beliefs about sexual violence | 0.306 *** | – | 0.306 *** |
| SDG-5 importance | −0.125 *** | – | −0.125 *** |
| Sex (women) | −0.012 | – | −0.012 |
| Age | −0.001 | | −0.001 |
| Educational level | −0.047 | – | −0.047 |
| Company size | −0.107 ** | – | −0.107 ** |
| **Prevention commitment** | | | |
| Implicit resistance | −0.174 * | – | −0.174 *** |
| Gender bias | −0.156 *** | −0.054 *** | −0.210 *** |
| Irrational beliefs about sexual violence | −0.071 | −0.053 *** | −0.124 ** |
| SDG-5 importance | 0.187 *** | 0.021 ** | 0.209 *** |
| Sex (women) | −0.039 | 0.002 | −0.036 |
| Age | −0.052 | −0.001 | −0.052 |
| Educational level | 0.109 ** | 0.008 | 0.117 ** |
| Company size | 0.016 | 0.018 * | 0.030 |
| **Prevention actions in the company** | | | |
| Implicit resistance | −0.207 *** | −0.018 * | −0.226 *** |
| Prevention commitment | 0.105 ** | – | 0.105 ** |
| Gender bias | 0.024 | −0.087 *** | −0.062 |
| Irrational beliefs about sexual violence | −0.039 | −0.076 *** | −0.116 ** |
| SDG-5 importance | −0.022 | 0.048 *** | 0.070 |
| Sex (women) | −0.051 | −0.001 | −0.052 |
| Age | −0.058 | −0.005 | −0.064 |
| Educational level | 0.021 | 0.022 * | 0.044 |
| Company size | 0.294 *** | 0.022 * | 0.317 *** |

Note: * $p < 0.05$; ** $p < 0.01$; *** $p < 0.001$. SEM using maximum likelihood with robust error corrections. Baseline vs. saturated likelihood ratio ($X^2 = 491.193$, $p < 0.001$). Bentler–Raykov $R^2$: overall ($R^2 = 43.99\%$), implicit resistance ($R^2 = 31.27\%$), commitment to prevention ($R^2 = 17.74\%$), and prevention actions in the company ($R^2 = 20.40\%$).

## 4. Discussion

This research provides the first empirical evidence of implicit managerial resistance towards preventing gender violence against women in organizations. The study is novel because, in addition to a conceptual framework, it provides evidence that managers' commitment to the prevention of gender violence against women is being affected by implicit resistance from the patriarchal culture, which is still in force.

The proposed conceptual model is feminist and is derived from the Social Dominance Theory (Sidanius et al. 2004) to the extent that it presupposes a resistance of the dominant power group to gender empowerment. In this regard, as early as the 1990s, the feminist movement warned about the emergence of "gender backlash" as a form of the social resistance of men towards women's advances in pursuing equal rights (Faludi 1991). In this context, resistance, as such, is foreseeable. In effect, this is translated in our finding that 3 out of 4 managers do not believe in violence against women because they consider it "biased" and think that only family or partner violence should be discussed, or that 2 out of 4 managers do not trust equality policies because they have "hidden interests". As

an individual indicator of resistance, 100% of managers report one of these views. As a composite resistance (the average of all of them), almost 1 in 2 managers have them.

Are these implicit resistances indicators of patriarchal culture? Indeed, the results show that behind the implicit resistance are gender biases and irrational beliefs that justify sexual violence. These variables are influential and explain 31.2% of the variation. Furthermore, as these increase, the resistances become more intense. Unfortunately, this result is statistically identical for both men and women. In this regard, although men have higher levels of gender bias than women, the level of implicit resistance is statistically similar. The possession of a managerial position may likely influence this lack of difference since female managers report less gender bias, less tolerance towards sexual violence, and more identification with SDG-5. In this regard, this result is consistent with the "queen bee syndrome," a phenomenon whereby—due to individual socialization and the patriarchal culture of organizations—women who hold high positions in organizations oppose feminist movements, are more critical of their female colleagues, and attribute their professional success to their own merits, preferring to surround themselves with more men than women (Xiong et al. 2022; Grangeiro et al. 2022).

The discovery of these resistances calls into question the high percentage of managers that say they are committed to or in support of prevention. Almost 9 out of the 10 surveyed managers report their favor; however, only 40% do not report intense resistance towards preventing gender violence against women. In other words, more than half of this "commitment" is undermined by resistance against it. Even in apparently "neutral" positions, 8 out of 10 managers have intense implicit resistance to prevention.

The implicit resistance shows that business involvement in prevention is a more complex process than previously thought and requires a reinforced strategy. Overcoming these resistances will not only encourage commitment to prevention but could also increase the effectiveness of prevention. Indeed, managerial attitudes towards gender violence are essential for organizational prevention. In Australia, for example, management has been found to play a central role in changing staff behavior (Hart et al. 2018); however, a high level of tolerance and social acceptance towards the perpetration of gender violence was also found, which limits its effectiveness. In another context, in a recent investigation of Bolivian companies (Vara-Horna 2022), it was found that the inequitable behavior of management is strongly correlated with high levels of workplace sexual harassment, violence against women, and tolerant attitudes towards violence in the staff. All of these variables were less prevalent in those companies where managers were more equitable.

Another significant result is that managers have less gender bias towards women (as individuals) than towards women within the organization. In other words, they recognize that women have more problems with violence and care obligations, but they do not acknowledge that they suffer from more labor discrimination, especially regarding promotion. In general, studies find that managers evaluate the climate of gender equality in their companies more positively than operational personnel, having a greater propensity to defend the status quo (Cortis et al. 2022). These findings question the effectiveness of leadership-only change strategies, where senior leaders are portrayed as effective agents of change for gender equality. These data are correlated with the fact that in this study, managers are more committed to preventing violence but are less committed to promoting gender equality. They better recognize the problem but have more difficulty recognizing the solution. The truth is that the substantial prevention of gender violence requires promoting gender equality in the organization. From what has been said previously, preventing gender violence requires specific training or protocols and a business structure where women have a relevant presence. In this regard, Dobbin and Kalev (2019, 2020) found that business training to reduce workplace sexual harassment is more effective in workplaces with more female managers since they are less likely to respond negatively to complaints and training.

Why is it essential to integrate the prevention of gender-based violence against women with the promotion of gender equality? Because doing the opposite can generate antagonism between the two. In other words, "prevention" could be used as new form of

discrimination against women. One of the consequences of initial prevention in organizations is "gender fatigue" (Kelan 2009), which is when organizations assume politically correct language, denying that discrimination exists, but, in practice, it persists. In this context, Thomas and Plaut (2008) warn that gender fatigue can reduce mentoring opportunities for women, affecting their professional growth. Indeed, Soklaridis et al. (2018) show that many men (in this case, doctors) avoid being alone with their female colleagues or subordinates to prevent "potential" situations of sexual harassment, in which they can be "unfairly" accused, thus reinforcing their bond and privileges with other men. Thus, apparent protection measures against workplace sexual harassment further reinforce gender discrimination and inequality.

Another relevant result is the protective role of valuing gender equality as a sustainable development objective. We found that when managers value gender equality, this value decreases implicit resistance and increases managers' commitment to prevention and preventive actions. Despite its protective role, only 1 in 3 managers consider it essential, being more prevalent in women than men.

### 4.1. Limitations

Although this study has addressed various sizes of companies and sectors, it is impossible to generalize the results to the case of microenterprises. The management and governance model of microenterprises requires independent research. However, it is assumed that due to the managers' relatively lower level of education and fewer available financial resources, the level of implicit resistance towards prevention may be much higher. Another limitation of this study is the high non-response rate. Many companies refused to participate, and of those that did participate, a significant percentage left the survey incomplete, making it invalid for analysis. Although these non-response rates are like those reported in studies that use digital surveys, it is also likely that the non-response rate is associated with an explicit rejection of preventing gender-based violence against women. This would mean that the results could reflect more intense resistances when incorporating larger samples.

### 4.2. Practical Implications

This study has practical implications. Most importantly, these results provide new content for managerial training in order to effectively involve managers in prevention. Each resistance detected, from the most passive to the most active, needs to be questioned and discussed with forceful arguments to favor compromise. Furthermore, this training should not exclude female managers since their resistance levels are statistically like those of their male colleagues. Finally, the training content should not only be aimed at recognizing the problem (gender violence against women) but also at the solution, which lies in promoting gender equality.

**Author Contributions:** Conceptualization, A.A.V.-H.; methodology, A.A.V.-H., Z.B.A.-G., L.Q.-C. and A.D.-R.; software, A.A.V.-H. and Z.B.A.-G.; validation, A.A.V.-H.; formal analysis, A.A.V.-H.; investigation, Z.B.A.-G., L.Q.-C. and A.D.-R.; resources, L.Q.-C.; data curation, A.A.V.-H. and Z.B.A.-G.; writing—original draft preparation, A.A.V.-H.; writing—review and editing, Z.B.A.-G., L.Q.-C. and A.D.-R.; visualization, A.A.V.-H.; supervision, Z.B.A.-G. and L.Q.-C.; project administration, L.Q.-C. and A.D.-R.; funding acquisition, A.A.V.-H. All authors have read and agreed to the published version of the manuscript.

**Funding:** This research has been financed by the European Union and the Spanish Cooperation, through the management of the Chamber of Commerce of Lima (CCL), within the framework of the project "Violencia de Género contra las Mujeres: fortalecer la prevención".

**Institutional Review Board Statement:** Ethical review and approval was waived for this study because it is not required for business research in Peru. However, all the recommendations of The Belmont Report (1979) were followed, ensuring informed consent, non-maleficence, and fairness.

**Informed Consent Statement:** Informed consent was obtained from all subjects involved in the study.

**Data Availability Statement:** The data presented in this study are available on request from the corresponding author.

**Conflicts of Interest:** The authors declare no conflict of interest.

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
