# Peer review of "Are Companies Committed to Preventing Gender Violence against Women? The Role of the Manager’s Implicit Resistance"

_socsci, doi:10.3390/socsci12010012_

Round 1

Reviewer 1 Report

This is an excellent paper. Concepts are well explained and the empirical model for measuring implicit resistance is a very significant contribution to the research on involving businesses in the prevention of gendered violence.

The paper would benefit from a rigorous edit to improve accessibility and clarity.

Author Response

Dear reviewer,

Thanks very much for the helpful and valuable comments on the article. All of them have contributed significantly to improving the document. As you can see, the general edition of the writing has been revised and improved thanks to the MDPI language service.

All the best, 

The authors. 

Reviewer 2 Report

Thank you for inviting me to review this work. First of all, I would like to thank and acknowledge the effort and work done by the authors in this study.

To follow and understand the comments made on your work, I inform you that I will respect the order and structure of your manuscript

The theoretical framework is very well argued and structured, I congratulate the authors for this.

I suggest reading the following current article that may help you to reinforce some of the concepts discussed and defined by the authors.

The reference is:

Barragán-Medero, F., & Pérez-Jorge, D. (2020). Combating homophobia, lesbophobia, biphobia and transphobia: A liberating and subversive educational alternative for desires. Heliyon6(10), e05225. https://doi.org/10.1016/j.heliyon.2020.e05225

When the authors state the objective in line 116, they state:

The research aims to demonstrate that managers' commitment to preventing gender… I consider it inappropriate to use the verb to demonstrate, in the social sphere little can be demonstrated given that reality is changeable and very subjective. I think it is more appropriate to use the verb to identify, to know...

Please correct this quote in the text. It is on line 556... power group to gender empowerment. In this regard, as early as the 1990s, Susan Faludi… Cite correctly, do not cite the author's name.

An important aspect that is not addressed if described in the study is the ethical aspects of the use and treatment of sensitive information. Did any ethics committee assess the study and its compliance with the Helsinki protocol?

How was the treatment and handling of the data carried out?

More information on these aspects should be provided.

The authors only indicated in lines 278 and 279 the following:

All participants were informed of the objective and nature of the study and guaranteed informed consent as well as confidentiality and anonymity. The questionnaire was digitized using SurveyMonkey.

Regards

Author Response

Dear reviewer,

Thanks very much for the helpful and valuable comments on the article. All of them have contributed significantly to improving the document.

  1. The general edition of the writing has been revised and improved thanks to the MDPI language service.
  2. Regarding the paper of Barragán-Medero and Pérez-Jorge, we have found the study very important for the new pedagogy against resistance towards sexual diversity. However, we have not found how to cite it in our research.
  3. The verb “to demonstrate” has been changed to “to evidence” in both the objectives and the abstract.
  4. We have also corrected the Susan Faludi quote.
  5. Regarding ethical aspects, we have added the following: “The present study follows a design that complies with the principles of ethics established in the Belmont Report (1979): respect, beneficence, and justice. Regarding the first principle, all participants were informed of the objective and nature of the study and guaranteed informed consent. Regarding non-maleficence, the research sought to preserve the personal safety of the respondents, giving it priority over information, through anonymous and confidential surveys and without individual access for the participating companies. The companies accessed a global report without identifying who responded in each case. Regarding the third principle of justice, the research results - globally - will be returned to the organizations with specific recommendations and dissemination events”.

All the best,

The authors.